# Carbon Quantum Dots Bridged TiO_2_/CdIn_2_S_4_ toward Photocatalytic Upgrading of Polycyclic Aromatic Hydrocarbons to Benzaldehyde

**DOI:** 10.3390/molecules27217292

**Published:** 2022-10-27

**Authors:** Jiangwei Zhang, Fei Yu, Xi Ke, He Yu, Peiyuan Guo, Lei Du, Menglong Zhang, Dongxiang Luo

**Affiliations:** 1Institute of Hydrogen Eergy for Carbon Peaking and Carbon Neutralization, School of Semiconductor Science and Technology, South China Normal University, Foshan 528225, China; 2School of Chemistry and Chemical Engineering, Huangpu Hydrogen Innovation Center, Guangzhou University, Guangzhou 510006, China; 3Great Bay Area Branch of Aerospace Information Research Institute, Chinese Academy of Sciences, Guangzhou 510530, China

**Keywords:** carbon quantum dots, photocatalysis, degradation, polycyclic aromatic hydrocarbons

## Abstract

Conversion of hazardous compounds to value-added chemicals using clean energy possesses massive industrial interest. This applies especially to the hazardous compounds that are frequently released in daily life. In this work, a S-scheme photocatalyst is optimized by rational loading of carbon quantum dots (CQDs) during the synthetic process. As a bridge, the presence of CQDs between TiO_2_ and CdIn_2_S_4_ improves the electron extraction from TiO_2_ and supports the charge transport in S-scheme. Thanks to this, the TiO_2_/CQDs/CdIn_2_S_4_ presents outstanding photoactivity in converting the polycyclic aromatic hydrocarbons (PAHs) released by cigarette to value-added benzaldehyde. The optimized photocatalyst performs 87.79% conversion rate and 72.76% selectivity in 1 h reaction under a simulated solar source, as confirmed by FT-IR and GC-MS. A combination of experiments and theoretical calculations are conducted to demonstrate the role of CQDs in TiO_2_/CQDs/CdIn_2_S_4_ toward photocatalysis.

## 1. Introduction

In fossil industrial societies, environmental pollution is closely related to the rapid progress of modern industry [1,2,3]. Among the kinds of pollutants, more and more public attention has been paid to the ecological environment change caused by organic pollutants [4,5]. In domestic sewage, organic pollutants usually include benzene compounds (Polychlorinated biphenyls) [6], formaldehyde [7], tobacco tar (PAHs) [8]. When tobacco and cigarette paper are insufficiently burned, the tobacco tars with polycyclic aromatic hydrocarbons (PAHs) as major pollutants may be formed [9]. PAHs are aromatic hydrocarbons with two or more fused aromatic rings, which can be combined in two forms, the non-viscous ring type and the viscous ring type [10]. It is well known that PAHs in the environment will be particularly harmful to human health [11], causing a variety of hazards to the respiratory system and circulatory system. Currently, the general methods of dealing with PAHs in pollutants include biological degradation [12], adsorption process [10] and photocatalytic degradation [13]. Although biodegradation is environmentally friendly, it can be restricted by various factors such as temperature, pH and microbial adaptations, and adsorption treatment prefers transferring organic pollutants to the adsorbent rather than reducing the concentration of it [14]. In contrast, semiconductor photocatalysis is considered as an appealing method for breaking down toxic compounds into non-toxic carbon dioxide and water under mild conditions [15]. Semiconductor-based photocatalytic reactions are environment friendly, having a high surface area, low cost and high modification, etc. [16,17]. It is worth noting that there is still little research on tobacco tar via photocatalytic degradation.

In recently years, abundant types of photocatalysts have been playing an important role, including ZnO [18], SrTiO_3_ [19], SnO_2_ [20], Bi_2_S_3_ [21], and so on. Unfortunately, the narrow absorption spectra and unsatisfactory photocatalytic performance limit their application. Thus, exploring highly active at the range of visible-light-driven photocatalysts is still a constant pursuit of researchers.

Titanium oxide (TiO_2_) is a classical semiconductor with favorable thermal and chemical stability [22,23]. However, limited by the wide band gap (3.2 eV) [24,25] and rapid photo-generated electron-hole recombination, the solar energy utilization of TiO_2_ is typically inefficient, resulting in a low photocatalytic efficiency [26,27]. Therefore, researchers have developed a variety of strategies for enhancing the photocatalytic performance of TiO_2_, such as precious metal deposition, pigment sensitization and metal ion doping, etc. [28]. Comparing with the above strategies, the construction of heterojunctions has been demonstrated to be an effective way for tuning the absorption and manipulating the charge separation [29]. For instance, a TiO_2_/CdIn_2_S_4_ hierarchical nano hetero-structure photocatalyst demonstrated an ~5.5 times higher in hydrogen production from water compared to bare TiO_2_ under visible light [30].

CdIn_2_S_4_, as a ternary metal chalcogenide compound, is equipped with an appropriate band gap (2.03–2.26 eV) and photoredox potential, promoting the light capture efficiency at visible light range [31,32]. Considering these advantages, CdIn_2_S_4_ has been reported to contribute to photocatalytic degradation of pollutants [33], water splitting [31], and carbon dioxide (CO_2_) photoreduction [34]. However, due to the photoinduced hole self-oxidation of bare CdIn_2_S_4_ and the rapid recombination of photoexcited charges, the practical application suffers a huge impediment [35]. In recent years, combining semiconductor materials with CdIn_2_S_4_ has been one of the approaches to solving the transfer direction of photogenerated carriers [36]. Moreover, the charge transfer rate of the composite photocatalyst system is closely related to the electrical conductivity between the material interfaces, which directly affects its catalytic activity [37]. Therefore, the introduction of materials with good electrical conductivity can facilitate the charge transfer of photoexcited semiconductors and co-catalysts.

Carbon quantum dots (CQDs), as a novel carbon nano material, are universally employed in various fields because of their excellent properties [38]. In the field of photocatalysis, CQDs have been used to modify binary semiconductor nano materials for its efficient electronic transmission capability [39]. For instance, Liang et al. showed a novel Z-scheme g-C_3_N_4_/CQDs/CdIn_2_S_4_ heterojunction showing an improvement of ~1.5 times higher photocatalytic degradation than g-C_3_N_4_/CdIn_2_S_4_ composite [40]. Pei et al. synthesized the ternary NiS/CQDs/ZnIn_2_S_4_ nanocomposite by a simple hydrothermal method. Compared with NiS/ZnIn_2_S_4_ heterojunction, the hydrogen production efficiency was more than 1.75 times [37]. Looking at these facts, it can be anticipated that CQDs play a crucial role in the charge transfer between TiO_2_ and CdIn_2_S_4_.

In this study, an active photocatalyst was developed by hydrothermal method that involves TiO_2_/CdIn_2_S_4_ heterojunction being modified with CQDs as bridge to improve the electron transport efficiency. Meanwhile, we investigated the effect of different ratios of CQDs on the photocatalytic performance of the TiO_2_/CQDs/CdIn_2_S_4_ heterojunction to obtain the optimum fabrication parameters. The composition, structure, morphology and optical properties of the prepared heterojunction were analyzed thoroughly. PAHs, a typical hazardous organic chemical in products of incomplete combustion tobacco, was chosen as the typical pollutant in tobacco tar. The photocatalytic degradation mechanism of PAHs by prepared heterojunction, the electron transfer and decomposition products were analyzed in detail.

## 2. Experimental

### 2.1. Chemicals

Thioacetamide (C_2_H_5_NS, Macklin^®^99% purity), indium nitrate hydrate, (InH_3_O_9_·xH_2_O, Aladdin^®^99% purity), cadmium nitrate tetrahydrate (CdN_2_O_6_·4H_2_O, Aladdin^®^99% purity), citric acid (C_6_H_8_O_7_, Macklin^®^98% purity), urea (CH_4_N_2_O, Aladdin^®^99.5% purity), acenaphthylene (ANY) (C_12_H_8_, Aladdin^®^97% purity), titanium oxide, rutile (60 nm) (TiO_2_, Macklin^®^99.8% purity) were used in this study.

### 2.2. Synthesis of CQDs Colloidal Solution

In a typical synthesis, 4 g of citric acid (AC) and 4 g of urea were added to 80 mL of distilled water and the prepared solution was stirred for 15 min at 600 rpm. Then, the mixture solution was transferred into a Teflon-lined cylinder in a stainless-steel autoclave for hydrothermal synthesis of CQDs at 120 ℃ for 12 h. The brownish-green final CQD colloidal solution was collected after cooling at room temperature naturally [41].

### 2.3. Preparation of the TiO_2_/CdIn_2_S_4_ Photocatalyst

The hydrothermal method was used to synthesize the TiO_2_/CdIn_2_S_4_ photocatalysts. The 80 mg TiO_2_ (60 nm), 0.308 g cadmium nitrate (Cd(NO_3_)_2_·4H_2_O), 0.600 g indium nitrate (In(NO_3_)_3_·3H_2_O) and 0.302 g thioacetamide (C_2_H_5_NS) were dissolved in 70 mL of distilled water. Then, the solution was taken to be stirred for about 4 h with the Teflon magneton. After that, the mixed solution was sent to a 100 mL Teflon-lined autoclave with hydrothermal method at 120 °C for 12 h. After cooling to room temperature, the as-prepared photocatalyst was washed with distilled water and ethanol three times and dried at 70 °C for 4 h.

### 2.4. Preparation of the TiO_2_/CQDs/CdIn_2_S_4_ Photocatalysts

The TiO_2_/CQDs/CdIn_2_S_4_ photocatalysts were prepared by the hydrothermal method. In detail, 0.154 g cadmium nitrate (Cd(NO_3_)_2_·4H_2_O), 0.300 g indium nitrate (In(NO_3_)_3_·3H_2_O) and 0.151 g thioacetamide (C_2_H_5_NS) were dissolved in 30 mL of distilled water (as solution A). Meanwhile, the CQDs dispersion divided into different gradients from 250uL to 2500 uL was dripped in 40 mL distilled water before 40 mg TiO_2_ was added and stirred for 1 h (as solution B). Then, solution A was slowly added into solution B to obtain solution C. After being stirred for 10 h, solution C was subjected to a 100 mL Teflon-lined autoclave and maintained at 120 °C for 12 h. The obtained final product was washed with distilled water and ethanol three times and dried at 70 °C for 4 h [30]. The scheme for the synthesis is presented in Figure 1.

### 2.5. Characterization

The crystalline phase and crystalline size of the samples were analyzed by using an X-ray powder diffraction (XRD) technique (Rigaku Corporation, Japan) with Cu Kα1 radiation (λ = 1.541866 Å), scanning angle from 10° to 80° (2θ). The morphology and structure of the TiO_2_/CQDs/CdIn_2_S_4_/photocatalysts were measured by transmission electron microscope (TEM, Japan-JEOL-JEM 2100F) and scanning election microscopy (SEM, American-FEI-Quanta FEG 250). Transmission electron microscope was employed for the elemental mapping (Japan-JEOL-JEM 2100 F). X-ray photoelectron spectroscopy (XPS) was measured by Thermo SCIENTIFIC ESCALAB 250Xi (ThermoFisher Nexsa, America, X-ray source-Al Kα ray (hν = 1486.8 eV) the work function: 4.97 eV). The FT-IR spectra in the range from 400 to 4000 cm^−1^ was analyzed by a thermogravimetric spectrum instrument (America-Agilent-7890). Electron paramagnetic resonance (EPR) measurements was obtained using a CW-EPR Bruker spectrometer (Germany Bruker-A200). Spectrum measurement system was applied to collect. The UV-vis absorption spectroscopy of the samples (DH-2000-BAL) and the steady-state photoluminescence (PL) spectra of the photocatalysts were recorded by fluorescence spectrometer (QE65000-FL). The gas chromatography-mass spectrometer (GC-MS) was used to analyze the photocatalytic degradation of PAHs. In this experiment, the temperature is programmed as follows: maintain the initial 45 °C for 1.5 min; 10 °C/min to 200 °C for 1 min; 2.5 °C/min to 250 °C for 1 min; 4 °C/min to 280 °C for 3 min; 5 °C/min to 300 °C and hold for 10 min.

### 2.6. Photocatalytic Experiment

The photocatalytic activities of TiO_2_/CQDs/CdIn_2_S_4_ (different ratio of CQDs), TiO_2_/CdIn_2_S_4_ and bare CdIn_2_S_4_ were studied by comparative absorption spectra, and the optimal proportion of CQDs in the photocatalyst was determined. The 50 mg of the as-prepared photocatalysts was added in the substrate solution. The 300 W Xe lamp was used as the light source to provide UV or visible light (long-pass filter (>400 nm)). To evaluate the photocatalytic performance for cigarette tar degradation, acenaphthylene (ANY) was employed for the photocatalytic experiments, due to its predominant content in cigarette tar. The substrate was dissolved in toluene solvent at 1 mg/L, and the amount of the photocatalyst was 0.5 g/L. Before the illumination, the mixture was stirred with Teflon magneton for 30 min under dark conditions to ensure uniform mixing of photocatalyst and polycyclic aromatic hydrocarbon molecules.

To calculate density functional theory (DFT) within the generalized gradient approximation (GGA) using the PBE [42] formulation, Vienna Ab Initio Package (VASP) was exploited [43,44]. In this study, we performed the projected augmented wave (PAW) potentials for describing the ionic cores and to account for valence electrons, working on 450 eV kinetic energy cutoffs [45,46]. With Gaussian smearing, partial occupancies of the Kohn–Sham orbitals were allowed at a width of 0.05 eV. The on-site corrections (DFT+U) were subjected to the 3D electron of Ti atoms (Ueff = 4.5 eV). Electronic energy was identified as self-consistent when the change in energy was less than 10^−5^ eV. The geometry optimization was known as convergent when the force change was less than 0.02 eV/Å. Grimme’s DFT-D3 approach was employed to illustrate the dispersion interactions.

We determined a = b = 3.858 Å and c = 9.652 Å for anatase TiO_2_ unit cells when employing a 10 × 10 × 4 Monkhorst-Pack k-point grid for Brillouin zone sampling. For Brillouin zone sampling, the equilibrium lattice constant of cubic CdIn_2_S_4_ is a = 10.920 Å, in the presence of a 2 × 2 × 2 Monkhorst-Pack k-point grid. Hexagonal graphene unit cells isolated by a vacuum layer of 15 Å depth were optimized for equilibrium lattice constants, being regarded as a = 2.468 Å, when employing a 15 × 15 × 15 Monkhorst-Pack k-point grid for Brillouin zone sampling. After that, two heterojunction surface models were constructed. The first was a heterojunction surface model based on TiO_2_/CdIn_2_S_4_ (101). The part of CdIn_2_S_4_ has a *p* (1 × 2) periodicity in the X and Y directions and one stoichiometric layer in the Z direction; the TiO_2_ (101) part has a *p* (1 × 6) periodicity in the X and Y directions and two stoichiometric layers in the Z direction; the TiO_2_/CdIn_2_S_4_ (101) slab was added with a vacuum layer in the Z direction at the depth of 15 Å in order to separate the surface slab from its periodic duplicates. The second heterojunction model was built by adding a graphene monolayer with a (4 × 5√3) periodicity between the two parts of model 1. During structural optimizations, the Γ point in the Brillouin zone was used for k-point sampling, and the bottom stoichiometric layer of the TiO_2_ (101) part was fixed while the rest were allowed to fully relax.

## 3. Result and Discussions

### 3.1. Characterizations of Structure and Morphology

The morphology and microstructure of the as-prepared photocatalysts were initially checked by scanning electron microscope (SEM). As shown in Figure 1A, TiO_2_ presented a spherical shape with uniform particle size of ca. 65 nm. After coupling TiO_2_ with CdIn_2_S_4_ through a hydrothermal method, the stacked nanospheres with wrinkles were monitored (Figure 1B), which is consistent with the previous reports [32,47]. The morphology can be attributed to that the anisotropy of CdIn_2_S_4_ drives a directional growth, leading to the presence of nanosheets and subsequent self-assembly. While, the decoration of carbon quantum dots (CQDs) on TiO_2_/CdIn_2_S_4_ resulted in no obvious morphological changes as observed by SEM, which can be explained by the small size of the CQDs (Figure 1C) [48].

To confirm the phase and crystal structures of the as-prepared samples, powder X-ray diffraction (PXRD) (Figure 1D and Appendix A) analysis was performed. On the one hand, the main diffraction peaks located at 2θ = 23.15°, 27.24°, 33.00°, 40.73°, 43.31° and 47.41° can be indexed to cubic crystal structure of CdIn_2_S_4_ (JCPDS#27-0060), corresponding to the indices of (220), (311), (400), (422), (511) and (440) planes, respectively [49,50]. On the other hand, the diffraction peaks in TiO_2_/CQDs/CdIn_2_S_4_ composite exhibit a standard phase of tetragonal TiO_2_ (JCPDS#21-1272) with the lattice (101), (004), (200), (105) and (211) [30]. In addition, CQDs (top right) showed a broad peak centered at 2θ = 22.57° with low intensity owing to the small size, which is consistent with the previous reports [48,51,52,53]. Thus, the (220) plane of CdIn_2_S_4_ at 2θ = 23.2° in TiO_2_/CQDs/CdIn_2_S_4_ did not alter significantly compared with TiO_2_/CdIn_2_S_4_ due to the low intensity of CQDs characteristic peak. The crystal size of CQDs is evaluated based on the PXRD patterns using Debye–Scherrer equation [53,54,55,56], which is capable of calculating the size of nanocrystals based on the X-ray diffraction features, regardless of the aggregation of these nanocrystals. Meanwhile, the full width at half maximum (FWHM) of as-prepared CQDs is measured to be 546.01×10^−4^ rad, suggesting an ultra-small size of ca. 2.5615 nm (Table 1), confirming that the synthesized samples were quantum dots with the unique property of quantum dots, whose size ranges from 1 to 10 nm. As seen in Figure 1D, in comparison to TiO_2_/CdIn_2_S_4_, TiO_2_/CQDs/CdIn_2_S_4_ presents no significant change, which is ascribed to the weak XRD signal of CQDs.

Transmission electron microscopy (TEM) was employed to further demonstrate the microscopic structure of TiO_2_/CQDs/CdIn_2_S_4_. The uniformly distributed lattice fringes in Figure 1E,F clearly present the intimate interfacial contact between TiO_2_ and CdIn_2_S_4_. The lattice spacings of 0.357 nm and 0.327 nm are exhibited in Figure 1F,G, which correspond to (101) planes of the tetragonal phase of TiO_2_ and (311) planes of the cubic phase of CdIn_2_S_4_ [33,57], respectively; this is in full accord with the PXRD results. Based on the crystal parameters of TiO_2_ and CdIn_2_S_4_, the lattice mismatch can be obtained as ∆d/dTiO_2_ = 8%. The small lattice mismatch typically indicates rapid transfer of photogenerated carriers between interfaces of TiO_2_/CdIn_2_S_4_ due to the lower interfacial trap state [58]. Next, energy dispersive spectra (EDS) mapping was employed to investigate the elemental composition and distribution of TiO_2_/CQDs/CdIn_2_S_4_ photocatalysts. As shown in Figure 1H, carbon can be clearly detected to be evenly distributed in the whole TiO_2_/CdIn_2_S_4_ region. In addition, Cd, In, S, Ti and O elements were also confirmed from the sample, which can correspond to the elemental composition of TiO_2_/CQDs/CdIn_2_S_4_ photocatalysts. For further confirming the content difference of CQDs between TiO_2_/CQDs/CdIn_2_S_4_ and TiO_2_/CdIn_2_S_4_, Appendix A show that C element was not found in TiO_2_/CdIn_2_S_4_ photocatalysts.

### 3.2. Band Structure of TiO_2_/CQDs/CdIn_2_S_4_ Photocatalysts

Figure 2A illustrates the UV-Vis diffuse reflectance spectrum (DRS) of bare TiO_2_, CdIn_2_S_4_, TiO_2_/CdIn_2_S_4_ and TiO_2_/CQDs/CdIn_2_S_4_ photocatalysts. The tetragonal TiO_2_ exhibited a typical absorption edges at 387 nm, situated in the absorption range of ultraviolet light [59]. Meanwhile, the absorption edge of CdIn_2_S_4_ can be observed at 610 nm corresponding to the band gap energy of 2.03 eV [49]. Furthermore, the loading of CdIn_2_S_4_ on TiO_2_ lead to a light absorption range of the composite extended to the visible region comparing to pristine TiO_2_ [30,60]. After the introduction of CQDs, TiO_2_/CQDs/CdIn_2_S_4_ presented a slightly enhanced absorption capability in range from 600 to 900 nm, which can be attributed to the narrow band gap energy of CQDs [48]. The valence state of the samples was evaluated by X-ray photoelectron spectroscopy (XPS), from which the characteristic binding energies of Cd (3d), In (3d), S (2p) and Ti (2p) were in good agreement with the rational valence state of the composite (Figure 2B–E). As shown in Figure 2F, the XPS profile of O 1s in the initial state presented three peaks located at 529.67, 521.97 and 533.02 eV, corresponding to the lattice oxygen, adsorbed oxygen [61] and surface hydroxyl, respectively [62,63,64]. Under UV illumination, we observed a dramatically reduced O 1s (533.50 eV) peak corresponding to the surface hydroxyl (Figure 2F). Considering the reductivity of hydroxyl species and that the valance of TiO_2_ was composed by O orbitals, this result indicates that the O in TiO_2_ featured highly active oxidation sites in TiO_2_/CQDs/CdIn_2_S_4_, possibly as a S-scheme structure (Appendix A) [60], where the hydroxyl species were oxidized by the photo-generated holes in the valence band of TiO_2_. Based on the valence plots depicted in the Appendix A, the valence band (E_VB_) of the CdIn_2_S_4_ and TiO_2_ were calculated to be 0.56 eV and 2.23 eV [65]. Meanwhile, due to the band gap energy (Eg) of the CdIn_2_S_4_ (Eg = 2.02 eV) and TiO_2_ (Eg = 3.20 eV) shown in Figure 2A, the band structure of the S-scheme of TiO_2_/CQDs/CdIn_2_S_4_ and type-II of TiO_2_/CdIn_2_S_4_ were constructed in Figure 2G. Notably, compared with the type-II, the photogenerated electrons in CB of TiO_2_ and the holes produced in the VB of CdIn_2_S_4_ are inclined to recombine on the CQDs bridge during the photocatalytic reaction of S-scheme under the Coulombic attraction between electrons and holes [66]. Since more holes are accumulated in the VB of TiO_2_, the oxidation ability of the S-scheme photocatalyst becomes stronger.

### 3.3. Influence of S Defects Formation

Figure 3A presents the cubic spinel structure CdIn_2_S_4_ indexing to the space group FD-3m (No.227), where the purple, magenta and yellow balls represent the Cd, In and S atoms, respectively [67]. The In atom is connected to six S atoms, forming the In-S octahedral structure. The blue and red balls represent the Ti and O atoms, which constitutes the tetragonal TiO_2_. The CQDs serve as a bridge to conduct electricity between TiO_2_ and CdIn_2_S_4_ (Figure 3B). To further investigate the roles of CQDs in TiO_2_/CQDs/CdIn_2_S_4_, the density of state (DOS) was conducted using first-principle calculations. As depicted in Figure 3C, the shallow trap states that primarily consisted of S (3p), O (2p) and In (5p) orbitals were projected, which are consistent with the pristine component (Appendix A). A peak corresponding to S 3p orbital at the Fermi level (E-E_f_ = 0 eV) indicates the presence of S defects [68]. The generation of surface sulfur vacancies typically leads to the accumulation of charges on adjacent Cd and In atoms, which may serve as highly active sites for intermediate chemisorption during photocatalytic degradation. Additionally, with respect to TiO_2_/CdIn_2_S_4_ (Figure 3C), after the introduction of CQDs (Figure 3D), the DOS of valence band orbital of O atom shows a clear downward trend due to the increased electron extraction from O, suggesting an enhancement in the oxidation capacity of TiO_2_ [69], which is consistent with our in-situ XPS result. Furthermore, in consideration of the critical role of radicals in the photocatalytic process, electron paramagnetic resonance (EPR) was applied to monitor the influence of CQDs on radical generation under illumination. The EPR results in Appendix A confirmed that the samples with/without are both capable of producing superoxide and hydroxyl radicals under illumination, and the radical concentration generated from the sample with CQDs is slightly higher than that of the control group.

### 3.4. Influence of CQDs Content on Oxidation Performance

The as-prepared photocatalysts were exploited to degrade PAHs from cigarette tar to produce valuable products. First, 10 mg of the as-prepared photocatalysts was added in PAHs substrate, then the reaction was conducted under 100 mW cm^−2^ simulated solar source with AM 1.5 G filter. A combination of gas chromatography-mass spectroscopy (GC-MS), Fourier-transform infrared spectroscopy (FT-IR) and UV-vis DRS were employed to evaluate the degradation rate and products. Before the photocatalytic tests, the PAHs substrate and photocatalysts were initially mixed and stirred under dark conditions for 30 min to achieve adsorption equilibrium, during which we monitored no obvious change in UV-vis DRS.

In terms of the photocatalytic performance, the ratio of CQDs in TiO_2_/CQDs/CdIn_2_S_4_ presented a strong impact on the photocatalytic degradation rate over PAHs (Figure 4A). Namely, the photocatalytic activity increased with the increased ratio of CQDs, it and reached the zenith at 20 to 30 mg/L, which was then followed with a deceased activity. This may be due to the overloading of CQDs leading to a reduced charge transfer efficiency between TiO_2_ and CdIn_2_S_4_, which suppresses the S-scheme features of the photocatalysts. To further optimize the ratio of CQDs, we tested the range of CQDs between 20 to 25 mg/L, in which the 23 mg/L presented a degradation rate up to 78.5% (Figure 4B). As illustrated in Figure 4C, the color of the substrate changed from the initial brown to a transparent colorless solvent in 1 h photocatalytic process. In addition to the loss of color, a blue shift of the absorption maximum (from 430 to 385 nm) was also monitored during the photocatalytic reaction (Figure 4D). As it is known that the molecular structure can be reflected by the absorption spectra, to be specific, a decreased number of conjugated carbon-carbon double bonds typically result in an absorption spectrum with a shorter wavelength [70]. In our case, the blue shift suggests that carbon–carbon double bonds in PAHs is possibly destroyed during the photocatalytic degradation. The comparison of the degradation performances among the serials of samples were exhibited in Figure 4E, particularly, the loading of rational content of CQDs between TiO_2_ and CdIn_2_S_4_ dramatically improved the photocatalytic activity, comparing to the control group (from 45.81 to 78.50%), suggesting the photoactivity can be efficiently promoted by adding CQDs. Furthermore, the PAHs degradation performance was compared as shown in Table 2. Besides, the stability of TiO_2_/CQDs/CdIn_2_S_4_ was checked by repeating the experiments 3 times using the recycled samples (Figure 4F) Comparing to the initial cycle, ca. 7.2% photoactivity loss was observed in the second cycle. Meanwhile, for further confirming the stability of photocatalysts in PAHs degradation, as shown in Appendix A, 97.6% samples were retained in the two TiO_2_/CQDs/CdIn_2_S_4_ sample groups after 5 h photocatalytic degradation. While in the third cycle, the photoactivity loss was less than 2%, relative to the second cycle. As confirmed by PXRD (Figure 4G,H), the characteristic patterns of TiO_2_/CQDs/CdIn_2_S_4_ basically remain the same after 3 reaction cycles. As depicted in Appendix A, the hot filtration test was conducted to confirm the compatibility of TiO_2_/CQDs/CdIn_2_S_4_ photocatalysts [71]. After 10 min of reaction under simulated sunlight (degradation of 65.2% PAHs), the photocatalyst was removed and filtered. Then the filtrate was illuminated for another 50 min under the same condition with no significant increase in degradation, confirming that no leaching of the photocatalyst components emerged [72]. Meanwhile, the ICP-MS analysis was performed to reveal the metal leaching of the pure filtrate [73]. The contents of Cd, In and Ti were 0.00055, 0.000554 and 0.05477 ppm, respectively, indicating the TiO_2_/CQDs/CdIn_2_S_4_ photocatalyst was stable and that the photocatalytic activity derives from the whole as-prepared photocatalyst, rather than from the constituents of possible leaching.

### 3.5. Photocatalytic Performance for PAHs

The obtained products were initially analyzed by FT-IR spectroscopy, as shown in Figure 5A. After degradation, the most obvious changes in FT-TR spectra are two newly appeared signals located at ca. 1260 and 1730 cm^−1^ with high intensity, which can be assigned to the C-C key stretching vibration and aldehyde group, respectively [79,81]. In addition, increased intensity of FT-IR signals were also observed at 2860 and 2930 cm^−1^, which can be indexed to the cyclohexane bands [82]. These results are in good agreement with the benzaldehyde (*m*/*z* = 106) and dimethylcyclohexane (*m*/*z* = 112) products recorded by GC-MS (Table 3) [83]. It should be noted that the conversion rates based on GC-MS are slightly higher than that of UV-vis DRS, since some side products might not be reflected by absorption spectra. In contrast to the conversion rate of 58.42% presented by the control group without CQDs, the degradation of PAHs using the optimal TiO_2_/CQDs/CdIn_2_S_4_, (Figure 5B) GC-MS monitored that 87.79% of PAHs was decomposed to benzaldehyde (63.88%), dimethylcyclohexane (21.18%) and side products (2.73%) of very low proportion after 1 h of reaction.

## 4. Conclusions

In summary, TiO_2_/CdIn_2_S_4_ with S-Scheme band structure was modified by the introduction of CQDs during the synthetic process. Based on the in-situ XPS tests and theoretical calculations, we found that the presence of CQDs between TiO_2_ and CdIn_2_S_4_ can efficiently promote the oxidation capability of the photocatalyst, which benefits the photocatalytic degradation of PAHs. Thanks to this, the TiO_2_/CQDs/CdIn_2_S_4_ presents outstanding photoactivity in converting the polycyclic aromatic hydrocarbons (PAHs) released by cigarettes to value-added benzaldehyde. The optimized photocatalyst performs 87.79% conversion rate and 72.76% selectivity in 1 h reaction under simulated solar source, as confirmed by FT-IR and GC-MS, which was more efficient than the control group (58.42% conversion rate).

## Data Availability

All data are available in the main text or the electronic Appendix A (ESI).

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
