# Peer review of "Carbon Quantum Dots Bridged TiO2/CdIn2S4 toward Photocatalytic Upgrading of Polycyclic Aromatic Hydrocarbons to Benzaldehyde"

_molecules, 2022, doi:10.3390/molecules27217292_

Round 1

Reviewer 1 Report

Authors designed a new photocatalyst for the oxidation of polycyclic aromatic hydrocarbons. The synthesized photocatalyst could be interesting for this reaction, but authors should improves their manuscript regarding these comments.

1. The introduction section is very long, in particular, the first paragraph contains a lot of general information. Please shrink this section.
2. Please add the TiO2 particle size to the chemical section.

3. Schemes of the catalyst synthesis and oxidation reaction could provide remarkable information with a glance. Please add two schemes as mentioned to your manuscript. Moreover, you can demonstrate the S-Scheme photocatalyst reaction mechanism there.

4.Authors used urea with citric acid to synthesis CQD, but I think you generated nitrogen doped CQD. Therefore, the nature of CQD needs to be elucidated by analysis like CHN elemental analysis or investigating nitrogen peak with focusing on XPS.

5. The particle size determination by XRD could not be correct for CQD since the peak at 22 appeared in both of the spectra and I do not think that this peak belongs to CQD.
6. please compare your results with the previous report and more highlight your photocatalyst advantages.

7. Authors did not discuss about stability of the catalyst. Please determine recyclability of the photocatalyst.

8. Please add the hot filtration test.
9. please add more references about TiO2 photocatalysts, including: a)Sci Rep. 2021, 11, 16148; b) RSC Adv., 2019, 9, 17129-17136; c) J. Phys. Chem. Solids, 2022, 161, 110434.

Reviewer 2 Report

In this work, Zhang J. et al. report the preparation of a quantum-dots-decorated TiO2/CdIn2S4 material that is used for the photocatalytic degradation of polycyclic aromatic hydrocarbons. Overall, while this kind of materials is not new, the work is generally well-carried out and well-presented and the authors make use of numerous techniques to characterize the materials and evaluate their performance which is quite active. Nevertheless, I feel there are some unclear points in this work that should go through major revisions and re-evaluation before being publishable. Detailed comments are provided below:

a) How do the authors prove that the CQD are really present in the CQD/TiO2/CdIn2S4 material? The carbon observed by SEM-EDX could come from contamination from adventitious carbon (XPS could be used to prove that). Is it possible to provide spectroscopic signatures (Raman, XPS, etc.) expected for the CQD? What is the difference in C content (%) by elemental analysis when CQD are added in the preparation and when they are not added?

b) I do not really agree with the authors interpretation of SEM images (Figure 1B) that "flower like nanospheres" were obtained. What I see are large microspheres of many different sized and I do not understand how the authors obtain nanometric sizes from the Debye-Scherrer equation. Possibly, the nanoparticles are aggregated and  need to be redispersed in a medium.

c) The O1s spectrum "before UV" in Figure 2 D shows a very large asymmetrical O(OH) signal at 532.34 eV. It is very unlikely that this peak can be fitted with a single component. In reality, in that region several signals are expected to be observed i.e. surface M-O species, M-OH, adsorbed water from moisture etc. (compare with: Journal of Industrial and Engineering Chemistry 2021, 104, 43-60) the authors should carry out a more careful deconvolution.

d) Figure 5b is not clear, which values refer to conversion and which to selectivity for each catalyst?

e) The authors analyze the reaction products by FT-IR but it would be much clearer to find which products are formed via 1H-NMR as each of the mentioned products has a specific spectral signature. The authors are invited to provided such kind of investigation and clarify the distribution of products.

f) A comparison of catalytic performance with other literature-published catalysts for the same process is recommended.

Round 2

Reviewer 1 Report

Unfortunately, authors did not responded the comments perfectly.

1- Scheme for the synthesis of photocatalyst should be added.

2- The explanation about the determination of particle size by XRD (comment 5 of the before revise) should be added to the manuscript as they explained in the response.

3- Authors should check hot filtration test and do that.

4- the references in comment 9 should be added to the manuscript. They said that we added but actually they did not.

Author Response

Dear editor,

  Thanks for your constructive comments concerning our manuscript entitled ' Carbon quantum dots bridged TiO2/CdIn2S4 toward photocatalytic upgrading of polycyclic aromatic hydrocarbons to benzaldehyde'. (molecules-1970602) Those comments are all valuable and very helpful for revising and improving our manuscript, and significantly guide our research. We have studied these comments carefully and made modifications which we hope meet with your requirement. Revised portions are marked in the manuscript. The main corrections in the manuscript and the responses to the reviewers' comments are listed below.

Sincerely,

Dongxiang Luo

[email protected]

Reviewer(s)' Comments to Author:

General comments: Unfortunately, authors did not responded the comments perfectly.

Comment 1: Scheme for the synthesis of photocatalyst should be added.

Fig. R1. Synthesis scheme for TiO2/CQDs/CdIn2S4 photocatalysts.

Response 1: Thank for your advice, the scheme of the synthesis of TiO2/CdIn2S4 photocatalyst has been added at the Scheme, as shown in Fig. R1.

Comment 2: The explanation about the determination of particle size by XRD (comment 5 of the before revise) should be added to the manuscript as they explained in the response.

Response 2: Thank for your advice, the explanation has been added to the submitted manuscript.

Comment 3: Authors should check hot filtration test and do that.

Fig. R2 The hot filtration test of TiO2/CQDs/CdIn2S4.

Response 3: I’m sorry for misunderstanding the meaning of hot filtration test before. As depicted in the Fig. R2, according to the reported method (J Hazard Mater, 2019, 364, 429-440ï¼›Optik, 2019, 192, 162943), after 10 minutes of reaction under simulated sunlight (degradation of 65.2 % PAHs), the photocatalyst was removed and filtered out of the reaction then the filtrate was illuminated for another 50 min under the same condition, showing no significant increase in degradation. And the ICP-MS analysis of the pure filtrate was used to measure the metal leaching of the photocatalyst. The contents of Cd, In and Ti were 0.00055, 0.000554 and 0.05477 ppm, respectively, indicating the photocatalytic activity emerges from the whole TiO2/CQDs/CdIn2S4 photocatalyst, rather than from the components of possible leaching and that the whole photocatalyst was stable.

Comment 4: The references in comment 9 should be added to the manuscript. They said that we added but actually they did not.

Response 4: I’m sorry for this stupid mistake. When I cited the references, the wrong version of the manuscript was used so that the submitted manuscript did not add the references above-mentioned. The references in comment 9 have been added at the [22], [25] and [27] of references, respectively, in the TiO2 part of introduction.

Reviewer 2 Report

I have reviewed the revised version of the manuscript and I found it strongly improved, so I suggest acceptance in current form.

Author Response

We greatly appreciate your help for our work.